# The Effects of Different Doses of ROCK Inhibitor, Antifreeze Protein III, and Boron Added to Semen Extender on Semen Freezeability of Ankara Bucks

**DOI:** 10.3390/molecules27228070

**Published:** 2022-11-21

**Authors:** Ömer Faruk Karaşör, Mustafa Numan Bucak, Mihai Cenariu, Mustafa Bodu, Mehmet Taşpınar, Filiz Taşpınar

**Affiliations:** 1General Directorate of Agricultural Research and Policies, Ministry of Agriculture and Forestry, Ankara 06800, Turkey; 2Faculty of Veterinary Medicine, Selçuk University, Konya 42003, Turkey; 3Veterinary Medicine Faculty, Agricultural Sciences and Veterinary Medicine University, 400372 Cluj-Napoca, Romania; 4Faculty of Medicine, Aksaray University, Aksaray 68100, Turkey

**Keywords:** Ankara buck, antifreeze protein III, boron, semen freezing, ROCK inhibitor

## Abstract

In the presented study, the effects of ROCK inhibitor Y-27632, antifreeze protein III, and boron at two different doses were investigated on the spermatological parameters of Ankara buck semen after freeze–thawing. Ejaculates were collected from bucks using an electroejaculator during the breeding season. The ejaculates that showed appropriate characteristics were pooled and used in the dilution and freezing of semen. The extender groups were formed by adding two different doses of three different additives (ROCK inhibitor Y-27632, 5 and 20 µM; antifreeze protein III, 1 and 4 µg/mL; boron, 0.25 and 1 mM) to the control extender. The semen was diluted with the different extenders at 35–37 °C and loaded into straws. Sperm samples frozen in liquid nitrogen vapors, following equilibration, were stored in liquid nitrogen. It was observed that extender supplementation improved post-thaw motility of Ankara buck semen after freeze–thawing. Differences were significant (*p* < 0.01) for 5 and 10 µM doses of ROCK inhibitor (71.82% and 74.04 % motility), as well as for 0.25 and 1 mM doses of boron (76.36% and 72.08% motility), compared to the control group (66.15% motility). With respect to the evaluation of acrosomal integrity and mitochondrial activity after freeze–thawing, although supplementation provided protection at all doses, the efficacy was not statistically significant (*p* > 0.05). It was observed that DNA damage was improved by antifreeze protein III at 1 µg/mL (1.23% ± 0.23%) and by boron at all doses (0.25 mM: 1.83% and 1 mM: 1.18%) compared to the control group (3.37%) (*p* < 0.01), following the thawing process. In the present study, it was determined that some additives added to the extender provided significant improvements in buck spermatozoa motility and DNA damage after thawing.

## 1. Introduction

According to data from the Food and Agriculture Organization (FAO), Ankara goats are a goat breed specialized for mohair yield. The Ankara (Angora) goat received its name from the town of Angora (now the city of Ankara) in the Central Anatolia Region, where it was first developed. Angora goat breeding in Turkey dates back to 2400 BC. They are bred for mohair production and brood yield to meet raw material needs, especially in the textile industry. The main countries of Angora goat breeding in the world are Turkey, South Africa, Argentina, the United States of America, Canada, New Zealand, Russia, and Brazil [1,2,3,4]. While the number of Angora goats in Turkey was around 2.5–3 million in the 1930s, it had decreased to 289,557 animals by the end of 2021 [4,5]. Therefore, to preserve their progeny, the germ cells must be kept frozen.

Studies of sperm cell freezing gained momentum after the discovery of the cryoprotective properties of glycerol by Polge et al. [6]. Many factors affect the success of freezing mammalian spermatozoa, such as extenders, some cryoprotectives, extension rate, freeze–thaw rates, and species-specific spermatozoon membrane structure [7,8,9,10]. For many years, studies of the freezing of buck semen have continued to improve the post-thaw parameters and fertility rates [8,11,12,13,14,15]. The most common medium for freezing goat semen is the Tris-based extender [8,11,13,16,17,18,19,20,21].

Spermatozoa have a fluid mosaic membrane structure consisting of lipids and proteins. The cell membrane has two main roles: protecting the membrane integrity/functional structure by separating the cells, and ensuring the exchange of various substances between the intracellular and extracellular environment. The main structures in the membrane are phospholipids, glycolipids, intramembrane proteins, and cholesterol [9,22,23]. The ability of these structures to remain intact and functional will determine the spermatological quality following thawing.

Spermatozoa undergo some changes in their chemical and physical structures during the freezing and thawing stages. The first change occurs due to the phase changes of cell membrane lipids during the cooling process of spermatozoa, as they cannot reorganize their membrane lipids against the temperature changes that occur during cryopreservation. The membrane lipids begin to undergo a phase change at a temperature range of 17 °C to 36 °C. This causes deterioration in the structures of proteins that regulate the membrane arrangement and ion metabolism. The second change that occurs involves crystal particles that begin to form in the extracellular regions at temperatures between −5 °C and −10 °C. During the formation of extracellular crystals, the intracellular space becomes supercooled. Cells undergo physiological and osmotic stress due to crystal formation, resulting in damaging effects [9,23,24,25]. During sperm thawing, small crystal particles in the cell dissolve and combine, that is, recrystallize and form larger crystal particles. Large crystal particles cause physical pressure on the cytoplasmic organelles and membrane, cut-like damage, and other anomalies [26,27]. Damage due to intracellular crystal formation and recrystallization has been demonstrated in hamster tissue culture cells [28] and turkey semen [29]. Damage to cell organelles during freezing and thawing results in decreases in spermatological parameters and fertility.

Two different reasons have been offered to explain the damage to DNA during sperm freezing and thawing: First, the theory proposed by Zribi et al. [30] is that the freezing procedure damages DNA, and that this damage occurs via caspase enzymes and apoptosis activation. Accordingly, apoptosis-like structures are formed in the sperm cells during the freezing process. Nuclear division, chromatin condensation, and mitochondrial expansion are among the apoptotic changes [31,32]. The second theory is the view that DNA damage results from the oxidative stress on the cell. It has been reported that nuclear DNA is damaged due to oxidative stress during the freeze–thaw process. Due to the small amount of DNA repair mechanisms, freeze–thawed semen is vulnerable to oxidative attack [33,34].

In order to reduce the effects of this damage, some additives are added to the extenders before freezing, in an attempt to improve spermatological parameters after the freeze–thawing process [11,18,21,35,36].

**Antifreeze proteins.** DeVries and Wohlschlag (1969) [37] observed that some specific proteins can increase the resistance of Antarctic fish to low temperatures and freezing, and these proteins have been accepted as antifreeze proteins (AFPs). AFPs are considered special proteins because they can change the size and shape of crystal particles and control their regeneration. They can be synthesized by many living organisms such as insects, fish, bacteria, and algae [38,39,40,41,42,43,44]. The effect of AFPs on freezing cells and tissues has been discussed by various researchers; it has been reported that they contribute positively to the maintenance of sperm, oocyte, and tissue-cell vitality [45,46,47,48,49,50,51].

**Rho-associated coiled-coil kinase (ROCK) inhibitors.** Rho kinases increase efficacy by participating in important physiological tasks, such as contraction of the cytoskeleton and smooth muscle, cell proliferation, adhesion, and apoptosis [52,53,54]. Two different ROCK isoforms, ROCK I and ROCK II, have been identified in mammalian cells. ROCK I is expressed in the lung, liver, testes, and sperm head and tail [55,56,57], while ROCK II is expressed in the brain and heart [58,59]. It has been reported that the formation of apoptotic bodies is prevented by the ROCK inhibitor Y-27632 [60]. In a study, it was observed that apoptosis in endothelial cells was reduced by ROCK inhibitor Y-27632 [61]. However, Y-27632 was also found to induce apoptosis by causing changes in actin filament integrity [62], and Y-27632 was found to have opposite interactions.

**Boron.** Since boron is used in many areas of industry, its importance is increasing day by day. Additionally, 73% of the world’s boron reserves are in Turkey [63]. Although many studies have been conducted on the toxicity of boron, it has not yet been found that it has any toxic effects on the human reproductive system [64,65]. There are two important hypotheses explaining the effects of boron. One hypothesis is that boron has a role in cell membrane function, stability, or structure, such that it influences the response to transmembrane signaling, influences transmembrane movement of regulatory cations or anions, or has a cryoprotective effect against cold-shock damage [66,67]. The other hypothesis is that boron is a negative regulator that influences several metabolic pathways by competitively inhibiting some key enzyme reactions [68]. The safe daily boron intake for adults has been expressed by the World Health Organization (WHO) as 1–13 mg/day [69]. It has been shown that daily 100, 200, and 400 mg boric acid supplementation can improve semen quality in rabbits and has a positive effect on physiological status [70]. Despite these positive effects, high doses of boron have been reported to have a toxic effect on reproduction and to cause testicular atrophy in male rodents [71,72]. In addition, studies on different animal species have reported that daily boron intake ranging above 25 mg/kg causes damage to spermatological parameters and testicular morphology [71,73].

The effects of ROCK inhibitors, antifreeze protein III, and boron on semen freezing have not been demonstrated in detail. For this reason, the aim of the presented study was to freeze Ankara buck semen in the presence of the above-mentioned additives, and thus to improve the spermatological parameters following the freeze–thawing process.

## 2. Material and Methods

### 2.1. Chemicals, Animals, and Sperm Freezing

The chemicals used in freezing the semen were as follows: Trizma, Sigma-Aldrich T6066; citric acid, Sigma-Aldrich C0706; fructose, Sigma-Aldrich F2543; glycerol, Sigma-Aldrich G2025; ROCK inhibitor, Cayman Y-27632; antifreeze protein III, A/F Protein Inc. Waltham, MA, USA); and boron, Fluka 15580 (Munich, Germany).

Within the scope of the study, ejaculate taken from 5 adult animals (3–4-year-old Ankara bucks) was used. The care, housing, and semen collection of bucks took place at the Selcuk University Veterinary Faculty Research and Application Farm. Ejaculate was collected by an electroejaculator three times a week for six weeks during the breeding season. The ejaculate samples with suitable characteristics (spermatozoa concentration ≥ 2 × 10^9^ spermatozoa/mL; motility ≥ 80%) were combined and included in the freezing process of semen.

The Tris-based extender solution consisting of 82.66 mM fructose, 96.32 mM citric acid, 297.58 mM Tris, 15% egg yolk, 1% penicillin–streptomycin–amphotericin B mixture, and 5% glycerol (pH values ranging from 6.8 to 7.2) was used for semen freezing. The pooled ejaculate was divided into seven aliquots, and each aliquot was diluted at 37 °C with one of the freezing extenders, containing either ROCK inhibitor (5 or 20 µM), antifreeze protein III (1 or 4 µg/mL), boron (0.25 or 1 mM), or no additives (control), for a total of seven experimental groups, to a final concentration of approximately 200 million spermatozoa per one straw (0.25 mL); the process took place in one step, using a 10 mL glass centrifuge tube. Diluted samples were aspirated into 0.25 mL French straws at room temperature, sealed with polyvinyl alcohol powder, and equilibrated at 5 °C for 2–3 h. After equilibration, the straws were frozen in liquid nitrogen vapors (~−100 °C) for 15 min and then plunged into liquid nitrogen for storage [11,36].

### 2.2. Post-Thaw Microscopic Sperm Parameters

This study consisted of 10 replications made during the breeding season. Samples with or without additives were evaluated in terms of spermatozoa motility, plasma membrane integrity, acrosomal integrity, mitochondrial activity, and DNA integrity, after freeze–thawing. For this, straws stored in liquid nitrogen for at least one week were thawed at 37 °C for 30 s.

Spermatozoa motility was evaluated by examining a drop of thawed and diluted semen sample placed between the slide and coverslip at 40× magnification under a phase-contrast microscope with a heating plate at 37 °C. The average motility in at least 3–5 microscope fields was recorded as the % motility ratio [74].

The LIVE/DEAD^TM^ Viability Kit (L 7011 Thermo Fisher, Waltham, MA, USA, SYBR-14/PI) was used to analyze sperm plasma membrane integrity. The freeze–thawed ram sperm samples were diluted with PBS at a ratio of 1:3. Next, 30 μL of the diluted sample was mixed with 3 μL of PI and 10 μL of FITC-PNA stock solutions. After adding the dyes, properly mixed ram semen samples were incubated at 37 °C for 10 min, and sperm activity was stopped by adding 1 μL of Hancock solution. A fixed volume of 2.5 μL of semen was delivered onto a microscope slide and then a fluorescence microscope (Leica DM 3000 Microsystems GmbH, Wetzlar, Germany; emission at 520 nm, excitation at 450–490 nm) was used to perform analysis of approximately 200 spermatozoa. Spermatozoa with green heads were considered to have intact membranes, and those with the red heads were considered to have damaged membranes (Figure 1). Spermatozoa plasma membrane integrity was expressed as % [75].

Acrosome integrity was assessed by modified fluorescein isothiocyanate conjugated peanut agglutinin (L7381 FITC-PNA, Sigma-Aldrich, St. Louis, MO, USA) and modified propidium iodide (PI) staining, as described by Nagy et al. [76]. The freeze–thawed ram sperm samples were diluted with PBS at a ratio of 1:3. After that, 60 μL of the diluted sample was mixed with 3 μL of PI and 10 μL of FITC-PNA stock solutions. The mixed sample was incubated at 37 °C for 10 min, 1 μL Hancock solution was added to block the sperm movement, and finally, 2.5 μL of the prepared sample was analyzed. Investigation of ram sperm acrosome integrity was performed under a fluorescence microscope at 400× magnification (emission at 520 nm, excitation at 450–490 nm, Leica DM 3000 Microsystems GmbH). Spermatozoa emitting green fluorescence were regarded as having damaged acrosomes, whereas cells displaying no fluorescence in the acrosome cap were considered to have undamaged, intact acrosomes (Figure 2). At least 200 cells were counted for each sample. Spermatozoa acrosome integrity was expressed as % [76].

The assessment of the mitochondrial membrane function of sperm was fulfilled using the JC-1/PI, as described by Garner et al. [77]. Ram sperm was thawed at 37 °C in a water bath for 30 s and diluted with PBS at a 1:3 ratio. Next, 300 µL of the diluted ram semen sample was gently mixed with 3 µL of PI and JC-1 stock solutions. The final mixture was incubated at 37 °C for 10 min, and 1 μL Hancock solution was added to the mixed sample to stop sperm movement. The assessment of mitochondrial membrane potential activity was performed by dropping 2.5 μL of the semen sample on a clean glass microscope slide and investigating approximately 200 spermatozoa under a fluorescence microscope (400× magnification). The stained orange or bright green fluorescence of the sperm midpiece was considered to indicate mitochondrial membrane potential activity, whereas the sperm midpiece with matte color indicated no mitochondrial membrane potential activity (Figure 3). Mitochondrial membrane activity of sperm was expressed as % [77].

The Comet assay (Single Cell Gel Electrophoresis Assay) was used to determine sperm DNA damage in each sample. The effect of DNA double-strand breaks on cell life is more effective and decisive than single-strand breaks. We chose to perform the neutral Comet assay because we wanted to detect the breaking effect of the substances on sperm cell DNA. The slides (26 × 76 mm) were covered with a thin agarose layer by using 1% normal melting agarose (NMA) (Sigma-Aldrich A9539). NMA (1%) was dissolved in PBS (Ca^2+^- and Mg^2+^-free) by heating in a microwave without boiling, and then cooling to 65 °C before covering the slides at room temperature. After the covering process, slides were allowed to dry at least overnight at room temperature before the assay. Low Melting Agarose (LMA) (Sigma A0701) was used to coat the sperm cells on the slide. LMA (0.75%) was dissolved in PBS by heating in a microwave without boiling and then used at 37 °C. Liquid-nitrogen-frozen sperm cells were kept in a water bath (37 °C) for 30 s, and after thawing, they were diluted with PBS (Ca^2+^- and Mg^2+^-free) at the appropriate rate. An aliquot of sperm cells was mixed (1:3) with melted LMA. Then, 18 µL of diluted sperm cells (approximately 100,000 sperm cells) was mixed with 54 µL of LMA and this mixture was spread on an NMA-coated slide. These slides were covered with a 24 × 60 mm coverslip and left on a cold metal plate to jellify for 20 min. The coverslips were removed, and the slides were incubated at +4 °C for one hour in the lysis solution (16.3 mmol/L TritonX-100 (Sigma T8787), 34.08 mmol/L N-Lauroylsarcosine sodium salt (Sigma L9150), and 8.97 mmol/L in DL-Dithiothreitol (DTT) (Vivantis, Shah Alam, Malaysia, PC0705) in 99 mL stock lysis solution). Stock lysis solution consisted of 2.50 mol/L NaCl (Sigma 31434), 99.93 mmol/L EDTA (Vivantis PC0706), and 9.90 mmol/L Trizma (Sigma T1503). Proteinase K (100 μg/mL) (Vivantis PC0712) was added to the lysis solution and mixed gently with a micropipette, after which the samples were incubated at 37 °C for 3 h. At the end of incubation, the slides were removed from the lysis solution and placed in an electrophoresis tank containing cold 1X TBE at +4 °C for 30 min. Electrophoresis was performed in TBE buffer (66 mmol/L Tris-base (Sigma T1503), 67 mmol/L Boric acid (Vivantis PR0607), and 1.5 mmol/L EDTA (Vivantis PC0706)) at 1 V/cm for 30 min. The slides were washed in 0.9% NaCl for 90 sec after electrophoresis and dried at room temperature for at least one hour. A solution that consisted of 0.9 mol/L trichloroacetic acid (Sigma 27242), 0.7 mol/L zinc sulfate heptahydrate (Merck 1.08883), and 68 mmol/L glycerol was used for fixation for 10 min. The slides were gently washed three times with deionized water and dried at room temperature [78,79]. Next, 50 µL (8 µg/mL) ethidium bromide solution (Vivantis PC0707) was dropped in 3 different places on each slide and covered with a coverslip. The slides were visualized under a fluorescence microscope (Zeiss, Jena, Germany, Axioscope 5). Sperm cells with and without comet formation were considered damaged and undamaged, respectively. Accordingly, sperm with comet formation observed in the Comet assay were considered damaged. Sperm without comet formation was scored as undamaged. The proportion of spermatozoa with damaged DNA was determined by counting at least 200 spermatozoa from each slide.

### 2.3. Statistical Analysis

Averages of data obtained from 10 different replications were used in the statistical analysis. The overall data were defined as mean ± SE. The means of microscopic sperm parameters were analyzed with Duncan’s post-hoc test and variance analysis to adjust considerable differences. Statistical analyses were verified through IBM SPSS (Version 22), and statistical relevance was adjusted at *p* < 0.01.

## 3. Results

The effects of ROCK inhibitor, antifreeze protein III and boron on parameters of post-thawed Ankara buck sperm were evaluated in seven independent experiments as shown in Figure 4 and Figure 5. Regarding the post-freeze–thawing motility evaluation, the addition of 5 and 10 µM of ROCK inhibitor (71.82% and 74.04%) as well as 0.25 and 1 mM of boron (76.36% and 72.08%) both showed better protection than that of the control group (66.15%) (*p* < 0.01). Although the use of antifreeze protein III at a dose of 1 µg/mL (70.58%) improved the motility of the control group (66.15%), it was not found to be statistically significant (*p* > 0.05).

When the sperm plasma membrane, acrosome integrity, and mitochondrial membrane activity were examined after freeze–thawing, it was determined that all doses of additives added to the extender failed to show significant effectiveness (*p* > 0.05).

When sperm DNA damage was examined after freeze–thawing (Figure 5), antifreeze protein III at 1 µg/mL (1.23%) and boron at all doses (0.25 mM: 1.83% and 1 mM: 1.18%) provided more cryoprotective effects compared to the control group (3.37%) (*p* < 0.01).

## 4. Discussion

In addition to its economic value, the Ankara goat represents an important part of Turkey’s cultural heritage. The Ankara goat’s most important yield is mohair. This product is seen as very valuable, also known as “diamond fiber” in the world, and it has an important place in the textile industry [3]. At this point, freezing the semen and inseminating the females at the appropriate time is a priority for the breeding and reproduction of Ankara goats. Therefore, minimizing the cold-shock damage that develops during freezing storage is the basic goal of successful freezing. The membranous structures in spermatozoa (plasma membrane, outer acrosomal membrane, and mitochondrial membrane) are extremely sensitive to the freeze–thaw process. Membrane structures are composed of two rows of phospholipid layers intercalated with proteins, glycoproteins, and glycolipids, and arranged in a fluid mosaic fashion. These structures are thermodynamic and consist of 65–70% unsaturated phospholipids. Those cause irreversible phase change and transition from the liquid phase to the gel phase because of membrane cooling. The resulting phase change leads to a change in the kinetics of intramembrane enzymes, reducing the post-thawed spermatological parameters [9,23,80,81,82,83]. 

In the present study, the effectiveness of ROCK inhibitor (Y-27632), antifreeze protein (AFP III), and boron at two different concentrations, which were thought to have positive effects on the reduction of cold-shock, osmotic, and oxidative stress damage that develop during freezing, were investigated on the spermatological parameters of Ankara buck semen after freeze–thawing.

Rho-associated coiled-coil kinase (ROCK) inhibitors are involved in the regulation of phosphorylation of proteins that form the cytoskeleton, such as myosin, actin, tubulin, and proteins associated with integral membrane proteins. Its role at this point is to take part in basic functions, such as contraction and rearrangement of the proteins that form the cytoskeleton [84]. In this study, it was observed that ROCK inhibitor increased the post-thawed motility more than in the control group. In a study conducted in cats [84], it was observed that the addition of ROCK inhibitor at high doses (40 µM) worsened sperm motility, as well as plasma membrane and acrosomal integrity, after freeze–thawing. The most effective protective dose was found to be 10 µM.

In a study conducted in pigs, it was reported that ROCK inhibitor adversely affects oocyte maturation at high doses [85]. In our study, it was observed that both doses of ROCK inhibitor insignificantly worsened the integrity of the plasma membrane and acrosome, while preserving mitochondrial activity. We can state that the reason for the different results in the studies is due to the differences in membrane structure and fluidity between species.

In a study performed in humans, inhibiting the ROCK signaling pathway preserved resistance to cold shock [86]. From the different results obtained, it can be concluded that the ROCK inhibitor has different effects depending on the dose and the differences in the membrane structure between species. The fact that there is very little work on germ cells with the ROCK inhibitor makes the issue important. It is also clear that promising dose-related results were obtained. For this reason, we can say that testing intermediate doses and conducting studies that will include fertility in different species will make great future contributions to the literature.

Antifreeze protein III provides protection against cold shock by preventing crystallization growth through binding to crystal structures with its hydrophobic feature during freezing [50,87,88]. Antifreeze proteins not only lower the freezing temperature, but also prevent the formation of large crystal particles associated with recrystallization during freeze–thawing [89]. It has been reported that antifreeze proteins bind to crystal formations with the hydroxyl and carbonyl groups, and protect membranes in a hypothermic environment by blocking ion channels and preventing ion leakage from the environment [45,90].

In the current study, it was determined that antifreeze protein III significantly preserved post-thaw motility (*p* < 0.05). The protective effects on acrosomal integrity and mitochondrial activity were insignificant (*p* > 0.05). In another study [91], it was observed that antifreeze protein III significantly decreased the post-thaw motility when used at a low dose (0.1 μg/mL), but approached the control group when used at a high dose (0.5 μg/mL). These results contrast with the current study. Lv et al. [92] showed that the use of low-dose antifreeze protein III (1 μg/mL) in bucks improved post-thaw motility, membrane and acrosome integrity, and mitochondrial activity [92]. This case seems to match the motility and membrane integrity results received from the current study, but conflict with the results regarding acrosome integrity. This strengthens the argument that antifreeze proteins exhibit different reactions at different doses and in different membrane environments.

Spermatozoa motility generally works depending on three main factors: regulation, structural integrity, and energy continuity. The regulation of movement is controlled in the middle part of the sperm, especially from the flagellar and axoneme regions. While the flagellar part provides motility, the principal part is responsible for hyper-activation [93]. However, unsaturated fatty acids and saturated protein channels in the middle part make this structure susceptible to free radical attacks. It is thought that the ROCK inhibitor, when added to the extender, creates a shield surrounding the midsection structures by wrapping the actin and myosin filaments to protect sperm motility and increase the resistance against free radicals. Spermatozoon motility requires highly active mitochondrial structures. The ability of these structures is dependent on the presence of ATP-based enzymes. The addition of additives with antioxidative properties to extenders contributes to the provision of ATP required for sperm motility by acting as a coenzyme in oxidative decarboxylation metabolism in mitochondrial activity, and causes a tendency to decrease the incidence of mitochondrial dysfunction in spermatozoon metabolism [94,95].

In this study, the protection of post-thawed sperm motility by adding ROCK inhibitor and antifreeze protein III to the medium can be explained by the above-mentioned mechanisms. It has been reported that doses above 1 µg/mL of antifreeze protein III worsened motility in rabbit semen after freezing and thawing [96]. This result supports the findings of our study.

In a study conducted on rabbits, it was emphasized that 100, 200, and 400 mg of boric acid added to the feed increased semen quality [70]. It has been reported that 40 ppm boron added to the feed of bucks improved sperm motility and quantity [97]. Hence, the fact that high doses of boron do not cause toxic effects on motility supports the results obtained from our current study. Despite the positive effects of boron, it has been reported that its use in high doses also causes toxic effects. In a study in mice, 2000 ppm boric acid added to the diet had no effect on testes, and a dose of 4500 ppm caused spermatological degeneration, loss of motility, and an increase in the rate of dead sperm from the seventh day of administration [73]. In rams, it was observed that 1 mM boron added to the Tris-based extender significantly improved the post-thaw parameters (motility and mitochondrial activity), while high doses (2 and 4 mM) worsened it. However, the therapeutic effect of low doses on spermatozoa viability and acrosome integrity, and the toxic effect of high doses, were not considered significant [98].

Sperm DNA damage, which is associated with low fertility and pregnancy rates, is an important criterion in the diagnosis of infertility and shows a positive correlation with spermatozoon parameters. It is also known that the freeze–thaw process causes an increase in DNA single/double-strand breaks and fragmentation. Free radical formation induced by cold-shock damage is also among the causes of DNA damage [99,100]. In a study, it was observed that 1 mM boron added to a Tris-based extender significantly improved the post-thawed DNA integrity of ram semen, while high doses (2 and 4 mM) worsened it [98]. Bucak et al. [21] showed that increasing doses of boron in ram semen frozen with a low glycerol extender did not induce any effect on DNA integrity according to the Comet assay, but showed a tendency to improve DNA fragmentation according to the TUNEL test [21]. In the current study, it was observed that 1 µg/mL of antifreeze protein III and two different doses of boron reduced DNA damage after thawing, and that high doses of additives showed a tendency toward increasing the damage. This case suggests that the cryoprotectant doses in extenders may influence the efficiency of boron. The reason for the lack of consistent correlations between parameters in studies with increasing or decreasing doses of boron is not understood.

Axonemes and fibrillar structures located in the middle of the spermatozoa and surrounded by mitochondria are responsible for sperm movement. The energy required for motility is produced by mitochondria by oxidative phosphorylation. Free species produced during the freezing and thawing of sperm cause axonemal damage and loss of motility because of ATP depletion [101,102]. In a study conducted in ram semen, it was observed that low doses of boron preserved mitochondrial activity in a low-glycerol environment [21]. In our study, we observed that boron addition preserved mitochondrial activity, although not to a significant level.

In another study, it was determined that 0.25 mM boron added to a ram semen extender improved post-solution sperm motility, plasma membrane and acrosome integrity, and mitochondrial activity. It has been emphasized that increasing doses of boron have a significant toxic effect compared to lower doses [21]. From this point of view, it can be concluded that boron can be used at an optimum dose of 0.25 mM or less in semen-freezing studies.

Ankara goats have always been appreciated for their mohair production. Thus, it is important to freeze the semen of valuable individuals and use it for artificial insemination at the appropriate time, for the formation of disease-free herds with high genetic merit. Successful freezing of semen depends on the improvement of post-thaw parameters.

Our aim was to improve the freeze–thawed sperm parameters of Ankara bucks. The semen of other species may differ in density, quantity, and sensitivity to freezing. Of course, it is necessary to carry out this study on the semen of other species and to ensure the effectiveness of the additives used. In this study, we added some additives to the extender before freezing Ankara goat semen, then estimated the effectiveness of those additives. It was determined that some of the additives added to the Tris-based extender (ROCK inhibitor 5 and 10 µM; boron 0.25 and 1 mM) provided a significant improvement in sperm motility while others, antifreeze protein III 1 µg/mL and boron (0.25 and 1 mM), decreased the damage to DNA.

## Figures and Tables

**Figure 1 molecules-27-08070-f001:**
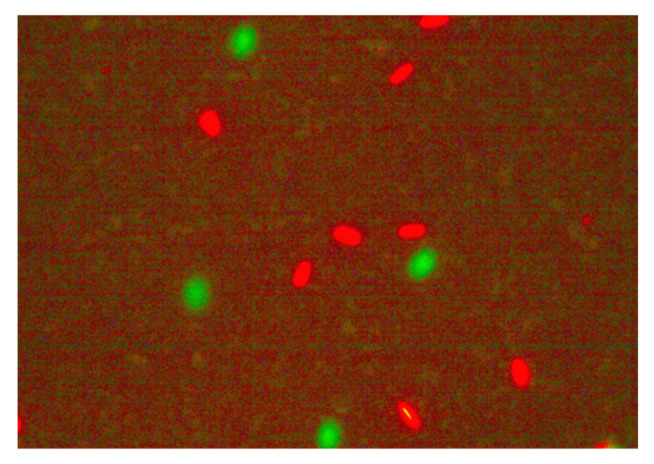
Spermatozoa with green heads were considered to have intact membranes, and spermatozoa with red heads were considered to have damaged membranes.

**Figure 2 molecules-27-08070-f002:**
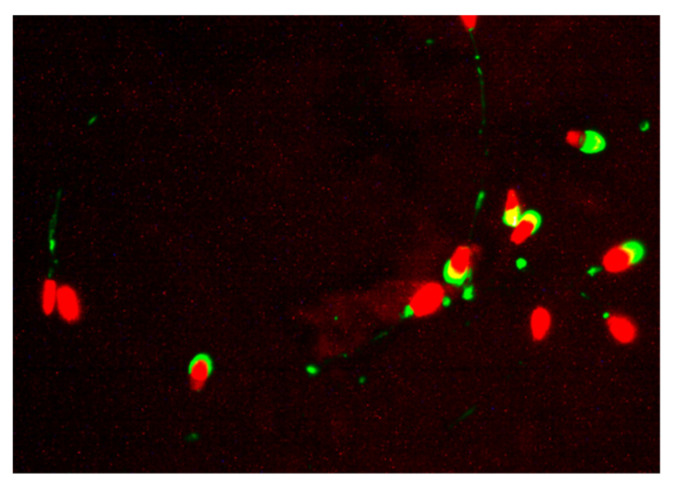
Spermatozoa emitting green fluorescence were regarded as having damaged acrosomes, whereas cells displaying no fluorescence in the acrosome cap were considered to have undamaged, intact acrosomes.

**Figure 3 molecules-27-08070-f003:**
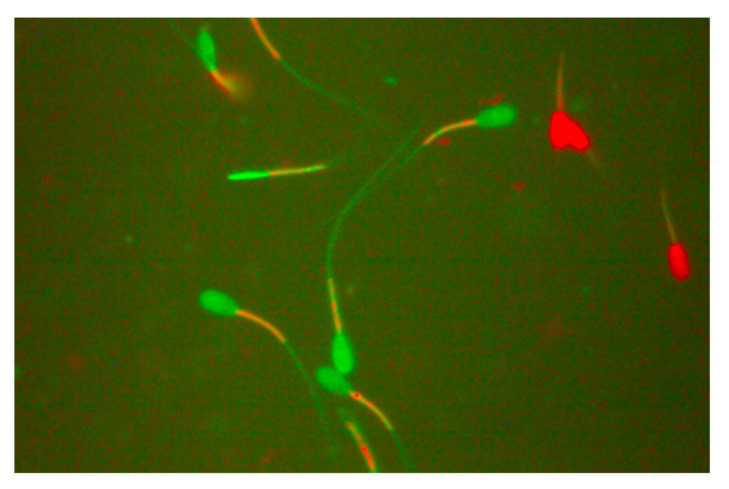
The stained orange or bright green fluorescence of the sperm midpiece was considered to indicate mitochondrial membrane potential activity, whereas the sperm midpiece with matte color indicated no mitochondrial membrane potential activity.

**Figure 4 molecules-27-08070-f004:**
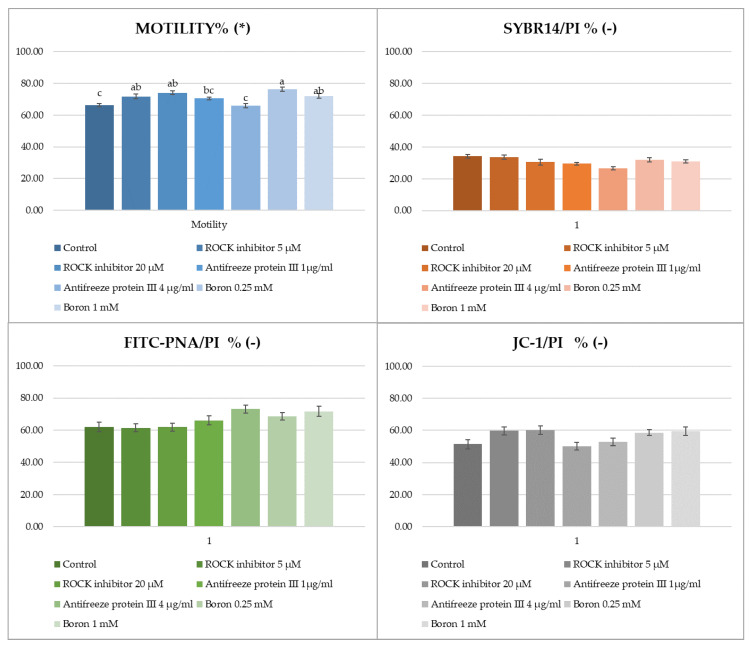
Findings of post-thawed motility, plasma membrane integrity (SYBR14/PI), acrosomal integrity (FITC-PNA/PI), and mitochondrial activity (JC-1/PI) (%) of Ankara buck semen frozen with different additives (± SH). (*): The letters a, b, c in the same column represents the statistical differences (*p* < 0.01). (-): Not significant (*p* > 0.05).

**Figure 5 molecules-27-08070-f005:**
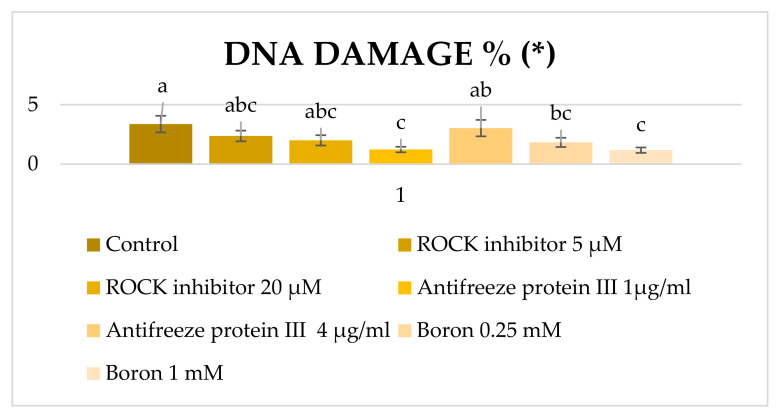
Findings of post-thawed DNA damage (%) of Ankara buck semen frozen with different additives (± SH). (*): The letters a, b, c in the same column represents the statistical differences (*p* < 0.01).

## Data Availability

The data presented in this study are available on request from the corresponding author.

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
