# Peer review of "The Effects of Different Doses of ROCK Inhibitor, Antifreeze Protein III, and Boron Added to Semen Extender on Semen Freezeability of Ankara Bucks"

_molecules, 2022, doi:10.3390/molecules27228070_

Round 1

Reviewer 1 Report

Please refer to the manuscript attached.

Author Response

We revised english grammar.

we revised ms to Reviewers view, showing them on ms.

Reviewer 2 Report

The manuscript is interesting and basically clearly written. Regardless of the factors relevant to the local industry, it brings significant data that may also be important for a wide audience in relation to the storage of reproductive material of various animal species. This significantly broadens the knowledge in this area. In my opinion, the manuscript may be published with a few minor corrections.

Manuscript points for correction:

The authors did not maintain consistency in reporting the results of the statistical analysis, which makes interpretation difficult. They considered p <0.01 as statistically significant, and p> 0.05 as statistically insignificant. It is not clear which group included the results with the coefficient p> 0.01 and p <0.05 (were statistically significant or not significant). This should be clarified.

In addition, due to the fact that the study concerned the examination of semen of only one species of animals, it is necessary to clearly indicate the weak points of the experiment, its limitations and the lack of universality of the obtained results. It is admittedly indicated in the discussion, but in my opinion it requires a clear separation in the text (as a separate point).

Author Response

Reviewer 2 revisions were performed and showed with red mark.

Reviewer 3 Report

Some minor issues have to be addressed before this manuscript is published.

1.      It will be easier to understand the results if the authors present them using figures instead of chart

2.      Representative images of each group are needed for the results shown here, for instance, JC-1/PI staining.

3.      There are grammar errors and typos in this manuscript, I list some of them here, but there are apparently more. Please go through the whole manuscript and carefully correct them.

1)      Introduction – Antifreeze Proteins – line 8: “maintainance” should be “maintenance”

2)      Discussion – Paragraph 7 – line 3: “activites” should be “activities”

3)      Discussion – Paragraph 12 – line 1: “spermatoza” should be “spermatozoa”

Author Response

Reviewer 3 revisions were performed and revised. English grammar revisions were made

Round 2

Reviewer 1 Report

Based on the review, the authors have made some amendments to the manuscript.